# Increasing the Hilbert space dimension using a single coupled molecular spin

Hugo Biard [1], Eufemio Moreno-Pineda [2], Mario Ruben [3,4,5], Edgar Bonet[1], Wolfgang Wernsdorfer [1,5,6 ✉] & Franck Balestro [1 ✉]

Quantum technologies are expected to introduce revolutionary changes in information processing in the near future. Nowadays, one of the main challenges is to be able to handle a large number of quantum bits (qubits), while preserving their quantum properties. Beyond the usual two-level encoding capacity of qubits, multi-level quantum systems are a promising way to extend and increase the amount of information that can be stored in the same number of quantum objects. Recent work (Kues et al. 2017), has shown the possibility to use devices based on photonic integrated circuits to entangle two qudits (with "d" being the number of available states). In the race to develop a mature quantum technology with real-world applications, many possible platforms are being investigated, including those that use photons, trapped ions, superconducting and silicon circuits and molecular magnets. In this work, we present the electronic read-out of a coupled molecular multi-level quantum systems, carried by a single $Tb_2Pc_3$ molecular magnet. Owning two magnetic centres, this molecular magnet architecture permits a 16 dimensions Hilbert space, opening the possibility of performing more complex quantum algorithms.

[1] CNRS, Grenoble INP, Institut Néel, Univ. Grenoble Alpes, Grenoble, France. [2] Depto. de Química-Física, Escuela de Química, Facultad de Ciencias Naturales, Exactas y Tecnología, Universidad de Panamá, Panamá, Panamá. [3] Institute of Nanotechnology (INT), Karlsruhe Institute of Technology (KIT), Eggenstein-Leopoldshafen, Germany. [4] Centre Européen de Sciences Quantiques (CESQ) within the Institut de Science et d'Ingénierie Supramoléculaires (ISIS), Strasbourg Cedex, France. [5] Institute for Quantum Materials and Technology (IQMT), Karlsruhe Institute of Technology (KIT), Eggenstein-Leopoldshafen, Germany. [6] Physikalisches Institut, Karlsruhe Institute of Technology, Karlsruhe, Germany. ✉email: wolfgang.wernsdorfer@kit.edu; franck.balestro@neel.cnrs.fr

For some decades, significant efforts have been invested in quantum information research, with the promise to revolutionise the way information is stored and processed. The strength of quantum computing lies in the possibility of using a coherent superposition of states, and interference between them, which enables a class of algorithms that are not accessible to classical computers. To achieve this goal, it is first mandatory to coherently manipulate and read-out a two-state system (qubit), as demonstrated in photonic devices[1], trapped ions[2,3], superconducting circuits[4], electronic spins in two-dimensional electron gases (2DEG)[5,6], phosphorous atom impurities in silicon[7–10], nitrogen-vacancy (NV) centres[11,12], or molecular systems[13,14], among others. The second step consists of the interconnection of $N$ entangled qubits, spanning the number of states to $2^N$, while preserving their quantum properties, in order to allow information processing, as demonstrated for example in superconducting circuits[15–17], two-dimensional electron gases[5,6], silicon devices[18] and magnetic molecules[19,20]. Towards the aim of scalability, the number of interconnected qubits has to be increased, as demonstrated by the 16-qubit processor of the IBM Q-experience[21], a 72-qubit quantum chip announced by Google[22], a 19-qubits computer tested by Rigetti[23], and a silicon spin-based 2-qubits processor obtained by QuTech[24]. More recently, even a NASA-Google partnership has claimed quantum supremacy[25]. Among these results, qudits, have been proposed as an alternative path towards the enlargement of the number of available quantum states for computation[26–29], as qudits allow to increase the number of states for the same number of quantum systems. Recently, the coupling of qudits has even been obtained using photons[30]. In their work, Kues et al. demonstrated an approach in which photons, possessing multiple high-purity frequency modes, are generated on an integrated micro-chip. With $d = 10$, they confirmed a 100-dimension Hilbert space, and validated this platform by measuring Bell inequality violations and performing quantum state tomography.

Towards this alternative, magnetic molecules possessing magnetic memory, better known as Single Molecule Magnets (SMMs), are a promising spin qudits platform[13,14,31]. These systems have the advantage of being synthesised in thousands of identical units and can be integrated in devices through bottom-up approaches, retaining their intrinsic magnetic properties[32–34]. SMMs can also be tailored through the use of different metal ions and ligands compositions for their integration in devices. Likewise, SMMs have been shown to exhibit quantum phenomenon such as Quantum Tunnelling of the Magnetisation (QTM)[35–38], quantum phase interference[39,40], and coherent spin manipulation with coherence times of the order of 1 μs[41]. This is the case of a bis(phthalocyaninato)terbium(III) SMM (TbPc$_2$), which has been integrated into transistor-like configurations[42–45] as well as in carbon-nanotube transistors[46,47]. Different studies have shown its magnetic resilience when deposited on surfaces, and the Tb$^{3+}$ ion redox stability[34]. In the TbPc$_2$, the π-electron system on the ligand eases the electric conduction, enabling electric read-out[42,44] of the nuclear spin $I = 3/2$ carried by the Tb$^{3+}$ ion. With coherence time of the order of 300 μs, the coherent manipulation of the different nuclear spin states was demonstrated, leading to the implementation of the Grover algorithm using this nuclear spin qudit[45]. This strategy clearly highlights the potential of qudits, as long coherence times and quantum algorithms have been achieved without requiring inter-qubit coupling. In this context, increasing the Hilbert space dimension is an appealing option for implementing more complex quantum algorithms or universal reversible quantum logic gates, such as the Fredkin or Toffoli gates.

Herein, in the quest of molecular spin systems with larger accessible Hilbert space, we present the first electrical read-out of the nuclear spin states, carried by a coupled multi-level system, achieved with a single-molecular unit of Tb$^{3+}$ triple-decker phthalocyanine, incorporated in a three-terminal transistor (Fig. 1a). We show that the magnetic moment of the two Tb$^{3+}$ ions effectively couples with the electronic structure of the Pc ligands allowing the electronic read-out of the nuclear spin states. More importantly, we observe the indirect coupling between the nuclear spin states residing on the Tb$^{3+}$ ions. We emphasise here that there is no entanglement between the two electronics or nuclear spins, but a larger Hilbert space dimension ($d = 16$) can be achieved via indirect interaction between the nuclear spins. Indeed, these results could represent an alternative for the realisation of more complex quantum algorithms using a single-molecular multi-level system.

## Results

**Tb$_2$Pc$_3$ single-molecule magnet.** to probe our coupling scheme, we chose a tris(phthalocyaninato)di(terbium(III)) Tb$_2$Pc$_3$ SMM, composed of two Tb$^{3+}$ ions and three phthalocyaninato (Pc) ligands. Both terbium ions sit on a distorted square antiprism geometry. As the TbPc$_2$ analogue, the total angular momentum $J = L + S = 6$ characterises the [Xe]$4f^8$ electronic shell of each Tb$^{3+}$ metal ion. Note that the strong spin-orbit coupling

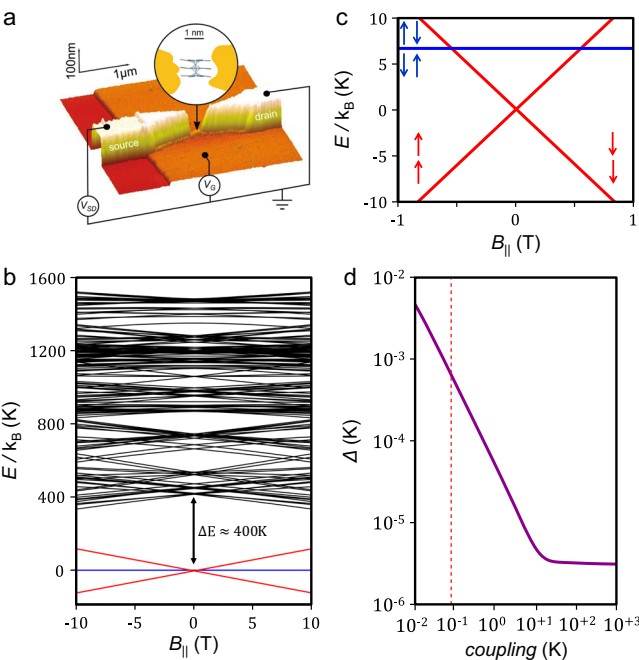

**Fig. 1 Geometry of the Tb$_2$Pc$_3$ single-molecule transistor and its electronic structure. a** AFM coloured image of Tb$_2$Pc$_3$ SMM connected to a gold source and drain electrodes through tunnelling barriers, on top of Au/HfO$_2$ gate (adapted from ref. [54]). The experimental set-up allows us to control the gate potential $V_G$ and the source-drain potential $V_{SD}$, while performing lock-in measurements through the latter. An enlarged picture of the Tb$_2$Pc$_3$ SMM is given in Supplementary Fig. 1. **b** Zeeman diagram of Tb$_2$Pc$_3$ SMM derived from Eq. (3) with the applied magnetic field **B** along the easy axis of the magnetisation the ferromagnetic and antiferromagnetic basic states are respectively represented in red and blue with separation of $\Delta E \approx 400$ K from the first excited state. **c** Enlargement of the ground ferromagnetic state and the first excited state (antiferromagnetic) un red and blue, respectively. **d** Evolution of the energy gap $\Delta_{m,m'}$ at the anti-crossing as a function of the coupling between the electronic spins of the two terbium ions. The red dashed line marks the $\Delta_{m,m'} = 0.6$ mK and the ferromagnetic coupling $C = 92$ mK, values employed for simulations (Supplementary Information).

separates the $J = 6$ state from the $J = 5$ by 2900 K, hence, at common operating temperatures, solely the $J = 6$ is considered. At a molecular level, the ligand field in which the $Tb^{3+}$ is located, created by the Pc ligands, leads to a doublet electronic ground state with strong uniaxial anisotropy. To calculate the Zeeman diagram for the $Tb_2Pc_3$ molecule, a ligand field Hamiltonian for $C_4$ geometry is expressed in the Stevens operators formalism:

$$H_{lf}^i = \langle r^2 \rangle u_2 A_2^0 O_2^0 + \langle r^4 \rangle u_4 (A_4^0 O_4^0 + A_4^4 O_4^4) + \langle r^6 \rangle u_6 (A_6^0 O_6^0 + A_6^4 O_6^4)$$

$$(1)$$

and the Zeeman Hamiltonian,

$$H_{Zeeman}^i = g_J \mu_0 \mu_B J_{||}^i \cdot B_{||}$$

$$(2)$$

where in (1) $u_i$ are the Stevens factors, $O_i^j$ are the Stevens operator and $A_i^j \langle r^j \rangle$ are the ligand field parameters. In (2) $g_J$, $\boldsymbol{J}$ and $\boldsymbol{B}_{||}$ are respectively the gyromagnetic ratio of $Tb^{3+}$, the electronic moment operator of $Tb^{3+}$ and the applied field vector. The presence of $O_i^j$ provides a strong anisotropy to $Tb_2Pc_3$, causing an energy gap of ca. 400 K between the $m_J = |\pm 6\rangle$ and the $m_J = |\pm 5\rangle$ electronic states. μ-SQUID[48] and ab-initio CASSCF[49] studies have revealed the anisotropy of the $m_J = |\pm 6\rangle$ doublet state of the $Tb^{3+}$ ions to have an easy axis perpendicular to the Pc planes.

Moreover, both $Tb^{3+}$ ions in the dimer are coupled through a ferromagnetic interaction, therefore, the Hamiltonian considering the ligand field and the interaction is:

$$H_{Tb_2} = H_{lf}^i + H_{Zeeman}^i - CJ_1J_2$$

$$(3)$$

where the first accounts for the ligand field for each $Tb^{3+}$ ion, the second term is the Zeeman term and the third term represents the coupling between the electronic spin of the $Tb^{3+}$ ions. Figure 1b shows the Zeeman diagram corresponding to the diagonalization of the $(2J + 1) \cdot (2J + 1)$ matrix. At zero field, the ferromagnetic ground state $|\pm 6, \pm 6\rangle$ and the antiferromagnetic excited state $|\pm 6, \mp 6\rangle$ are separated from the first excited state $|\pm 5, \pm 5\rangle$ by $\Delta E \approx 400$ K (Fig. 1b). Note that at our experimental electronic temperatures ($\approx 80$ mK) only the $m_J = |\pm 6, \pm 6\rangle$ is populated.

The presence of the transverse terms $O_4^4$ and $O_6^4$ in the ligand field Hamiltonian induces mixing of the $|+6, +6\rangle$ and $|-6, -6\rangle$ states via $J_4^\pm$, resulting in an opening of the anti-crossing between the $|\pm 6, \pm 6\rangle$ states. At very low temperatures, this opening acts as a relaxation pathway for the temperature-independent relaxation of the magnetisation, i.e., QTM process. At the anti-crossings the tunnelling probability is governed by the Landau–Zener equation:

$$P_{m,m'} = 1 - \exp\left(-\frac{\pi \Delta_{m,m'}^2}{2\hbar g_J \mu_B |m - m'| \mu_0 dH_{||}/dt}\right)$$

$$(4)$$

where $\mu_0$ is the vacuum permeability, $\mu_B$ is the Bohr magneton, $m$ and $m'$ are the two states involved ($|m - m'| = 24$ for $Tb_2Pc_3$) and $\Delta_{m,m'}^2$ is the anti-crossing splitting energy, which is contingent upon the coupling between the $J_1$ and $J_2$ states. Calculation of $\Delta_{m,m'}^2$ for different ferromagnetic coupling values $C$ is presented in Fig. 1d. The red dashed line represents the crossing between the ferromagnetic and antiferromagnetic states[48]. From this value, we obtain $\Delta_{m,m'} = 0.6$ mK. This large value of the anti-crossing energy results in a Landau–Zener probability $P_{m,m'} = 1$ even for magnetic field sweeping rates of $10^3$ T/s. As the maximum magnetic field sweeping rate accessible for the experiment was lower than 100 mT/s, reversal of the magnetisation is only governed by QTM, and no direct relaxation from the ferromagnetic state to the antiferromagnetic one was observed.

Up to this point, solely the electronic states of $Tb_2Pc_3$ have been considered. From now on, we consider the nuclear spin $I = 3/2$ carried by each $Tb^{3+}$ ions. Considering the strong hyperfine interaction, each $|\pm 6\rangle$ ground state is split into four states unevenly separated corresponding to $m_I = \pm 3/2, \pm 1/2$. The effect of the nuclear spins and quadrupolar interaction is considered in the following Hamiltonian:

$$H_{hf}^i = A_{hf} I^i J^i + P(I^i)^2$$

$$(5)$$

where $A = 30.9$ mK and $P = 14.4$ mK are the hyperfine interaction and the quadrupolar constants respectively, considered to be the same for both $Tb^{3+}$ sites[48]. As nuclear spins have magnetic moments ~2000 times smaller than electronic spins, direct nuclear spin interaction can be safely neglected. In contrast, the nuclear spins can be indirectly coupled via the dipolar interaction between the $Tb^{3+}$ ions, resulting in $4 \cdot 4 = 16$ possible nuclear spin states. Hamiltonian in Eq. (6) accounts for all interactions in $Tb_2Pc_3$.

$$H_{Tb_2} = H_{lf}^i + H_{Zeeman}^i + H_{hf}^i - CJ_1J_2$$

$$(6)$$

Numerical diagonalization of the Hamiltonian in Eq. (6) results in a ferromagnetic ground state with multiple level-crossings in the range $B_{||} = \pm 46$ mT (Fig. 2a). The square regions in Fig. 2b indicate the anti-crossings. In these regions, electronic spins reversal is allowed via QTM, while preserving the nuclear spin states. As calculated, a total of 10 anti-crossings allow the reversal of the electronic spins. Nevertheless, at three of these positions the electronic spins reversal occurs at the same magnetic field $B_{||}$, therefore, only seven positions in the magnetic field where QTM occurs should be observed.

**Molecular spin states read-out.** the nuclear spin states read-out of the $Tb_2Pc_3$ molecular magnet was studied via transport measurements through the molecule while sweeping the external magnetic field in two directions. The quantum dot is defined by the Pc ligands which are known to be good electrical conductors due to their π delocalised links. Moreover, the overlap between the 4f shell of the $Tb^{3+}$ ions and the π delocalised links of the Pc ligands give rise to an exchange coupling[44]. Our detection scheme is then based on an exchange interaction between the electronic spin of the $Tb^{3+}$ ions and the nearby Pc read-out quantum dot. The coupling between the two systems results in a spin-dependent conductance through the read-out dot and establishes an all-electrical single spin detection. Sweeping back and forth the external magnetic field $B_X$ in the direction of the coil, the differential conductance is simultaneously recorded using a digital lock-in amplifier. Additionally, a constant external magnetic field $B_Y$ is applied along the other direction of the coil. Considering a single magnetic field sweep, we observe an abrupt jump in the differential conductance as presented in Fig. 3a. This abrupt variation is due to the reversal of the electronic spins carried by the two $Tb^{3+}$ ions[42–47,50]. Sweeping $B_X$ back and forth, a hysteresis is clearly observed in Fig. 3a. By measuring the difference of the differential conductance $\Delta(dI/dV)$ while sweeping back and forth the magnetic field $B_X$, for different values of $B_Y$, we observe different regions where abrupt jumps are measured (Fig. 3a). From now on, we will focus on the magnetic field region highlighted by the green double arrow in Fig. 3b. Other regions correspond to other molecular systems for which the nuclear spin signatures were not resolved.

Figure 3c presents the histogram compiling the differential conductance jump positions for 6000 back and forth magnetic field $B_X$ sweeps at 50 mT/s with an additional constant $B_Y = 1$ T. Note that we were not able to sweep the external magnetic field in the direction of the easy axis of the $Tb_2Pc_3$ molecular system due to a missing coil in the Z-direction of our experimental setup. As presented in Fig. 3c, two main features are observed: (i) peaks in

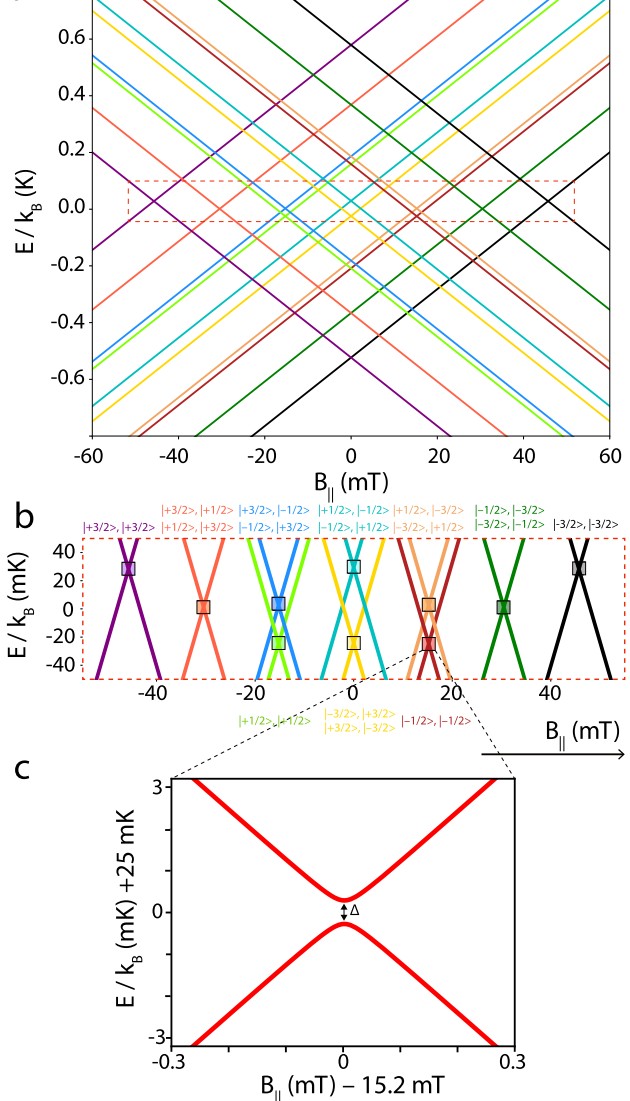

**Fig. 2 Zeeman diagram of the ground ferromagnetic state of Tb$_2$Pc$_3$.**
**a** Tb$_2$Pc$_3$ Zeeman diagram, considering all interactions. The ten colours are associated with the sixteen coupled nuclear spin states. Note that some of the events are degenerated. **b** Enlargement highlighting the ten anti-crossings, where the electronic spins reverse simultaneously. For each of the ten anti-crossings, the electronic spins have a Landau–Zener probability $P$ to reverse from a $|+6, +6\rangle$ state to a $|-6, -6, \rangle$ state. **c** Enlargement of one of the anti-crossings, with an anti-crossing energy $\Delta$.

the histogram with a high amplitude and roughly equally spaced, (ii) peaks in the histogram with a smaller amplitude and width and unequally spaced from the previous ones.

(i) We first focus on the main feature corresponding to the peaks in the histogram with a high amplitude and relatively large width (counts framed in grey in Fig. 3c). Seven peaks are observed, corresponding to the ten anti-crossings where tunnelling is allowed as expected from the theoretical predictions. Note here that the experimental values do not correspond to the QTM positions expected from Eq. (6); consequence of the miss-alignment of the easy axis of the Tb$_2$Pc$_3$ molecule with our applied magnetic field. As a result, while the positions of the QTM in a parallel magnetic field are expected within $B_\parallel = \pm 46$ mT, the measured positions using an external magnetic field occur within $B_X = \pm 55$ mT.

(ii) Two additional peaks are clearly visible at magnetic field $B_X$ higher than $+55$ mT and lower than $-55$ mT. First, the amplitude of these two peaks is lower compared to the amplitude of the main seven peaks. Moreover, the magnetic field spacing is lower than the spacing obtained for the main seven peaks. Therefore, these additional peaks in the histogram could be attributed to signatures of another molecular system we were not able to distinguish. This phenomenon will need further investigations and is a probable source of the large envelope observed in Fig. 3c, d. It is important to state that additional peaks should also be present in between those two, but the amplitude is too low to be observable. However, in the following studies, quantitative measurements could be approximative, as an accurate determination of the different nuclear spin states population would require not to take into account these additional peaks events, with a much lower amplitude, that could not be discriminated from the main seven peaks.

We now present the study of the probability of the QTM events considering the states degeneracy and the positions of the anti-crossings. For both coupled Tb$^{3+}$ ions, a total of 16 states is expected. As described earlier, out of these 16 states, 10 are not degenerate, while three of the anti-crossings occur at the same magnetic field. As a result, a total of seven states are truly observable. The nuclear spins states distribution can be obtained considering the 16 states as follows: the two most external peaks at $\pm 55$ mT (positions #1 and #7) have a probability $P = 1/16$ each, while the two immediate neighbours at $\pm 38$ mT (positions #2 and #6) and $\pm 20$ mT (positions #3 and #5) account for $P = 2/16$ and $P = 3/16$ respectively. Finally, the events at zero magnetic field (position #4) have $P = 4/16$, which corresponds to the double electronic spin reversal for 4 different nuclear spins. The results are summarised in Table 1.

**Coupled nuclear spins dynamics.** The dynamic of the nuclear spins was studied analysing the same 6000 back and forth magnetic field sweeps. A correlation cartography of the identified abrupt conductance jump positions is presented in Fig. 4a. The abscissa corresponds to the reversal of the electronic spins during the field sweep, while the ordinate corresponds to a reversal during the next sweep. The white dashed lines indicate the anti-crossing positions for which events are predicted. In order to ease the reading, a Gaussian filter is used to bring out the density of the local events. The utility of this correlation cartography is twofold: (i) first we obtain a better visualisation of the nuclear spins states; (ii) we can visualise to what extent the quantum system remains in the same nuclear spins state in between two sweeps through the anti-crossing regions.

Using the correlation cartography, it is possible to obtain an estimation of the lifetime of the nuclear spin states compared to the measurement time ($\approx 3$ s). As the two axes correspond to the trace and retrace $B_X$ magnetic field sweeps, two subsequent measurements with the same nuclear spin states are situated on the diagonal of the correlation map, whereas the off-diagonal positions correspond to nuclear spin state relaxation or excitation. As presented in Fig. 4, the correlation cartography does not exhibit an excess of events on the diagonal. It reveals a nuclear spins states lifetime shorter than the measurement time. Moreover, taking into account that the seven anti-crossing regions have different degeneracies, we renormalized the event counts as per the cumulated degeneracy of the corresponding abscissa and ordinate anti-crossings. The result is presented in the histogram Fig. 4b. We observe a higher population of the up left and bottom right blocks, clearly demonstrating the quantum system relaxation in between two consecutive measurements.

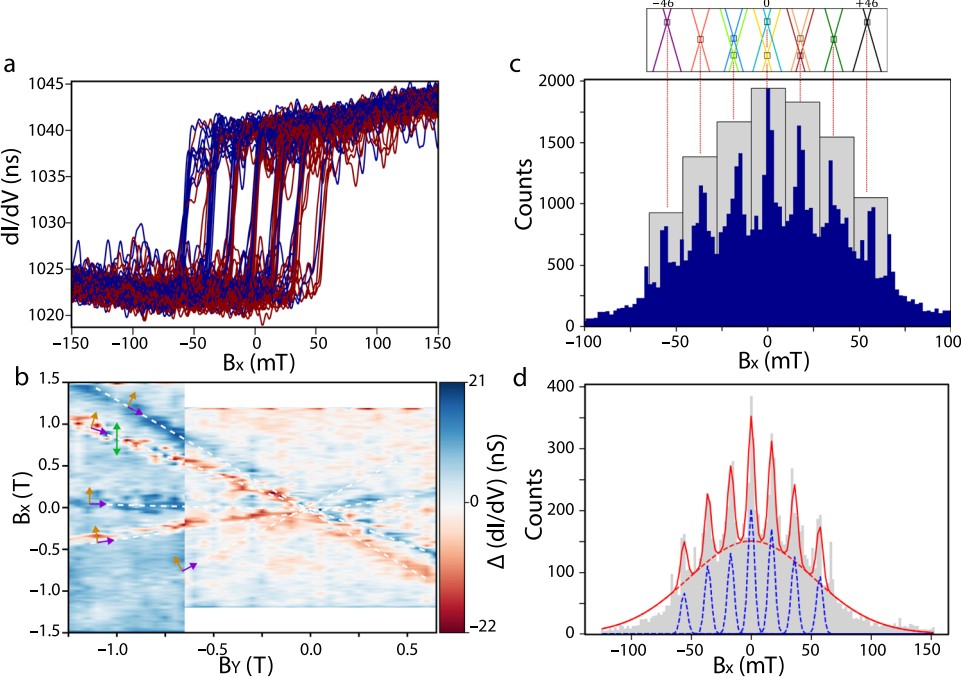

**Fig. 3 Electronic read-out of the coupled spin. a** Selected differential conductance traces for back-and-forth sweeps highlighting the seven expected tunnelling transitions. **b** Magnetic hysteresis map as a function of the applied field. The gate voltage $V_G$ is fixed and the source-drain voltage $V_{SD}$ is fixed to zero. For each value $B_Y$, the differential conductance measured during the back-and-forth magnetic field sweeps are subtracted, and then plotted. Several coloured lines are observed, corresponding to the hysteresis zones of each anisotropic magnetic moment, evolving linearly as a function of the transverse field. The green arrow indicates the studied region. The other regions were also studied but the nuclear spin states could not be resolved. **c** Histogram of the conductance jumps detected during 6000 magnetic field sweeps. The panel above the Zeeman diagram shows the predicted crossings (enlarged by 20%), showing the correspondence with the experimental observations. **d** The population histogram is fitted by seven Gaussian of the same width but different amplitudes, for the seven identified peaks; and another one with different width and amplitude for the envelope of undetermined events. To ease the interpretation, the jump positions in the histograms have been centred at zero as regard to the central peak.

**Table 1 Theoretical and experimental QTM probability for Tb₂Pc₃.**

| Position | #1 | #2 | #3 | #4 | #5 | #6 | #7 |
|---|---|---|---|---|---|---|---|
| Theoretical/% | 6.25 | 12.50 | 18.75 | 25.00 | 18.75 | 12.50 | 6.25 |
| Experimental/% | 8.95 | 13.38 | 16.13 | 18.78 | 17.67 | 14.95 | 10.14 |

As a result, the population distribution of these peaks is in good agreement with the relative degeneracy of each expected QTM magnetic field position in the Zeeman diagram. Assuming that all coupled nuclear spins are equally populated, the anti-crossing regions do not have all the same population.

A more detailed insight into the effective temperature of the nuclear spins states has been obtained by analysing the population of the states for 6000 forward magnetic field (sweeps from negative to positive magnetic field values). We present in Fig. 5a the renormalisation of the populations of the different nuclear spins states as per each anti-crossing degeneracy. Due to a low lifetime compared to the measurement time, the occupancy of states exhibits a relaxation towards a thermal equilibrium, as shown by the measurement of a higher population for the nuclear spin states with lower energies. To obtain a quantitative analysis of the temperature of the nuclear spins, we used Maxwell–Boltzmann statistics:

$$\frac{N_i}{N} = \frac{g_i e^{\frac{-E_i}{k_B T}}}{Z(T)} \text{ with } Z(T) = \sum_{i=1}^{n} e^{\frac{-E_i}{k_B T}}$$

where $Z(T)$ is the system partition function, $k_B$ is the Boltzmann constant, $T$ is the temperature, $E_i$ are the energies separation compared to the lowest energy level, and $g_i$ their respective degeneracies. $N_i$ is the population of each state, with the total population $N = \sum_{i=1}^{n} N_i$. Note that $N_i/g_i$ is the normalisation used previously for the correlation cartography. From this study, we obtain an effective temperature $T \approx 290$ mK of the nuclear spins, which is consistent with the one obtained on a TbPc₂ molecular magnet, measured with a comparable experimental set-up. We believe that the short lifetimes could be a result of flow current through the Tb₂Pc₃ unit, which induces a larger FWHM, and thus decreases the fidelity of the system[50].

Herein we have presented the electrical transport measurements of a Tb₂Pc₃ molecule embedded in a transistor, resulting in the first electronic read-out of a single coupled molecular spin. The read-out procedure consists of the detection of abrupt conductance jumps induced by the QTM of the electronic spins, in agreement with theoretical predictions. The seven measured QTM positions in magnetic fields are a direct consequence of the indirect coupling of the nuclear states residing in each Tb³⁺ ion via the electronic interaction. Moreover, we estimated a low lifetime compared to our measurement time which has been observed and ascribed to the current flowing through the molecular unit[50]. These two aspects can be certainly enhanced by device design. Our results clearly demonstrated the potential of coupled multi-level systems to span the Hilbert space using a single-molecular magnet, an important characteristic for the realisation of more complex quantum algorithms. Furthermore, the uneven separation of the different coupled nuclear spins would in principle allow the manipulation of each individual coupled state. These two characteristics may path the way to the realisation of more complex algorithms such as Fredkin or Toffoli.

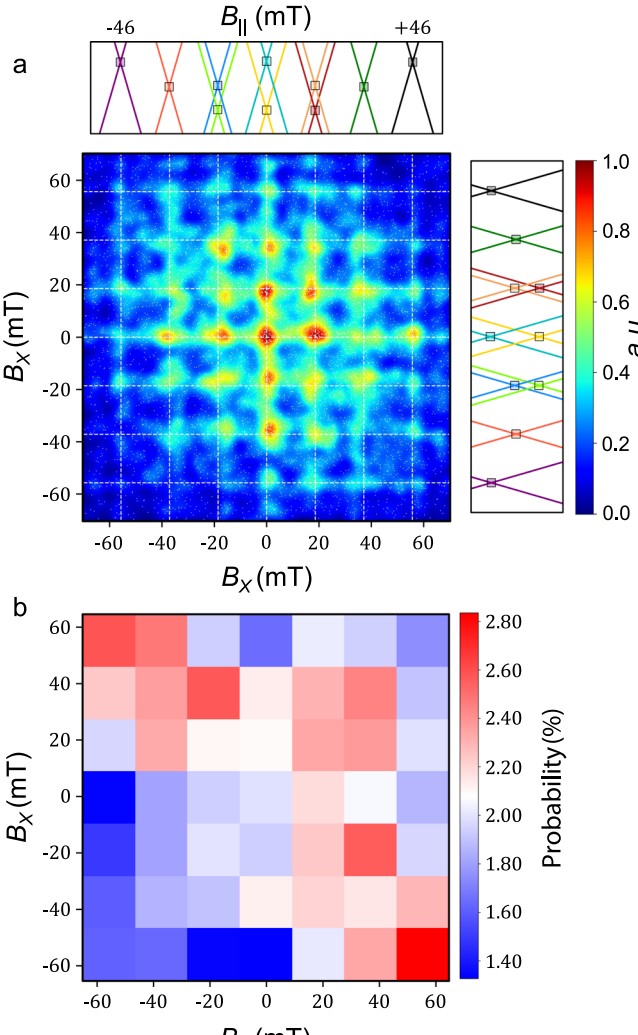

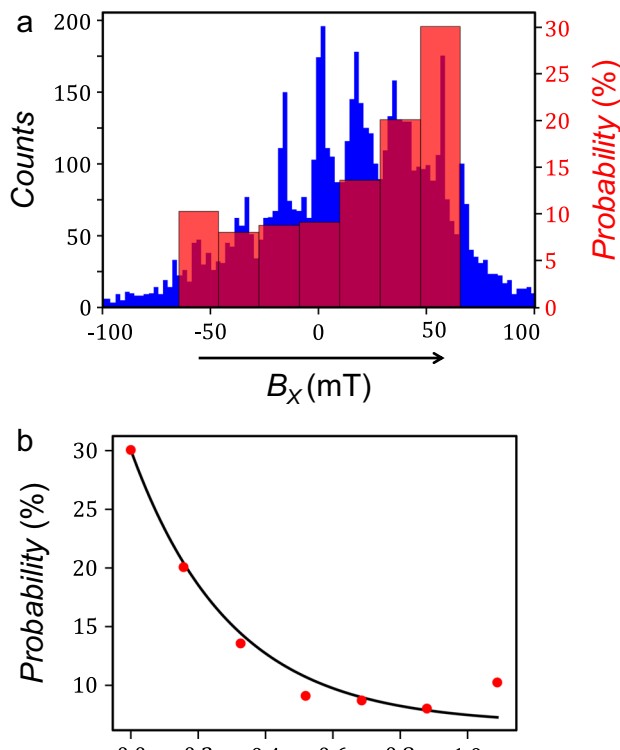

**Fig. 5 Effective temperature of the nuclear spins. a** Histogram of the electronic spins reversals position for 6000 magnetic sweeps. The field was swept from negative to positive values only. **b** Interpolation of the seven renormalized populations by a Maxwell–Boltzmann distribution. The extracted effective temperature is $T \approx 290$ mK.

**Fig. 4 Transition matrix of the QTM events. a** Correlation measurement of the electronic spins reversals. The abscissa corresponds to a conductance jump position during a magnetic field sweep, representing the reversal of the electronic spin orientation, and the ordinate to the conductance jump position during the next sweep. A Gaussian filter is applied to highlight the higher densities. The Zeeman diagram is rescaled to ease the identification of the anti-crossing positions, where QTM is predicted. **b** Correlation measurement histogram of the electronic spins, taking into account the anti-crossing degeneracies. The events are binned into a 7 × 7 histogram and normalised by the sum of the corresponding anti-crossing degeneracies, thus expressed as a percentage composition. The blocks corresponding to a nuclear spins state change exhibit a higher probability.

the magnetisation reversal measurements were performed in a 3He/4He dilution refrigerator so the sample temperature was below 100 mK. The cryostat used for this work was equipped with two coils, generating up to 5.5 and 1.25 T with a maximum sweep rate up to 100 mT/s, and allows the sample to rotate with respect to them, providing a 3D control of the applied magnetisation. A more detailed description of the sample fabrication and experimental set-up can be found in Chaps. 4 and 5 in ref. [53].

## Data availability
The data that supports the findings of this study are available from the corresponding authors upon request.

## Methods
**Nanofabrication**. Gold nano-constrictions acting as source and drain electrodes are fabricated through e-beam angles depositions, on the top of a gate composed of a metallic gold part and a high-k hafnium dioxide. It has to be noticed that the obtained electrodes arrow-like geometry enables to reduce the screening effect of the gate electric field[51]. The molecules were diluted in a solvent and later deposited at room temperature after a chemical cleaning (reactive-ion etching) of the sample surface. Afterwards, the electromigration process was achieved at low temperatures (4 K). The electromigration process consists of applying a difference of potential in between two electrodes forming a nano-constriction, so that the conduction electrons put in motion the gold ions, until a nano-interstice is opened[52]. In order to control the interstice size, the potential difference has to be precisely controlled with a real-time electronic. Thereby, when the conductance is below the so-called quantum of conductance ($G_0 = 2e^2/h$), the process is stopped in around 10 µs.

**Cryogenic measurement**. If the first conduction measurements between the source and drain electrodes were showing nano-metre size quantum dot signatures,

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

## Acknowledgements

We thank E. Eyraud, C. Thirion, T. Crozes and L. Abbassi for help with cryogenic set-up, maintenance, data collection and treatment, and sample preparation. We also thank S. Lumetti, M. Nale, M. Urdampilleta and J. Viennot for helpful discussions. H. B. kindly thanks the 2015 AGIR-POLE-PEM programme for PhD funding. We acknowl-edge the DFG-TR 88 "3Met" (project A8) and the Karlsruhe Nano Micro Facility (KNMF, www.kit.edu/knmf) for the provision of access to instruments at their labora-tories. EMP thanks the Panamanian National System of Investigators (SNI, SENACYT) for support. W.W. thanks the A. v. Humboldt foundation and the ERC grant MoQuOS No. 741276.

## Author contributions

H.B. prepared all samples, conducted the measurements and analysed the data. F.B., M.R. and W.W. conceived the idea and supervised the project. The data were discussed and analysed by H.B., E.B., F.B and W.W. E.B. created the software to conduct the mea-surements in the device. The synthesis and bulk characterisation of the $Tb_2Pc_3$ molecule was conducted by E.M.-P. and M.R. The manuscript was written by H.B., E.M.-P. and F.B. and discussed and analysed by all authors.

## Funding

## Competing interests

The authors declare no competing interests.
