## [Peer Review File · Nature Communications]

Reviewers' Comments:

Reviewer #1:

Remarks to the Author:

The manuscript entitled "Increasing the Hilbert space dimension using a coupled molecular spin qudit" reports on electrical transport measurements on a Tb₂Pc₃ molecule embedded in a transistor, as the first electronic read-out of two coupled qudits. The read-out procedure is based on the highly efficient detection of abrupt jumps in the differential conductance at the 'anti-crossing' field positions where QTM of the electronic spins occur.

One of the innovative aspects of the study is the demonstration through measurement, of the possibility to expand the Hilbert space of quantum objects by embedding dimers of qudits within the same molecule, here a Tb₂Pc₃ molecular magnet. The strategy essentially exploits the possibility of reading electronuclear transitions associated with weakly coupled lanthanide ions that carry magnetically active nuclear spins. The quadrupole term causes an uneven separation between hyperfine states, which together with the expansion of the Hilbert space, opens up new possibilities for individual qubit read-out and complex logic operations.

From my point of view this paper is sufficiently innovative to be published in Nat. Commun. and will benefit the molecular magnetism community and others involved in quantum information research.

The authors probed unambiguously that interaction between nuclear and electronic states occur, and even quantified the dipolar coupling between qudits. Data agree well with the theoretical calculations and the experiments reported are appropriately described.

Reviewer #2:

Remarks to the Author:

The manuscript of Biard et al. reports on proof-of-concept experiments that demonstrate the electrical readout of nuclear spin states of a prototypical molecular dimer. More specifically experiments were performed on a single "triple decker" Tb₂Pc₃ molecule embedded between electrodes in a transistor geometry. The key idea is novel and experiments are very challenging, thus results are unique and very interesting and they will certainly impact different fields and communities (molecular magnetism, quantum technologies, molecular electronics, spintronics etc.) and inspire several groups. In my opinion, this work merits publication in some way and it fits well the standard and audience of Nature Communication.

Said that, I find two main weak points that need to be clarified and corrected.

1. The magnetic field plays a key role in the whole work/manuscript but several discrepancies are evident comparing plots and reading the text. First, different orientations are not clearly defined: "B_x is the direction of the coil (axis?) ...while B_y is applied along the other direction of the coil (?)" but to account for the Zeeman energy, B_z is introduced to indicate the molecular axis which is obviously randomly oriented with respect to the coil (x,y). Later (for instance in fig.1), B_z is also named as B_{||}. Secondly, the absolute value of the field spans different values: the level crossing is expected in (0-60)mT range (see fig.2, fig3a and c; fig.4 and 5) but a static field of about 1T is applied along x- and y- directions to bias the system to QTM regime. Authors honestly notice that: "... the experimental values do not correspond to the QTM position expected by eq.6..." and they also pointed out that in these experiments the system missed one orientation of the applied field, yet, as matter of fact, the presentation of results is not clear at all and there is probably something wrong.

2. From the very beginning (eg in the title title, abstract, introduction) as well as in the conclusions, the molecular spin system presented as promising qudit offering a large Hilbert space but it is not clear whether the authors look for a large multilevel quantum system or for two entangled qudits. In the first case, no correlation (entanglement) is needed among the states (but only different energy spacing between levels) in the second case the entanglement between two distinguishable units play a key role. As a matter of fact, the Tb₂Pc₃ molecule presents two permanently coupled qudits but the entanglement between states is not discussed in this work.

Although this does not affect much the results -which are focused only on the readout-, it sounds odd and even misleading for readers of Quantum Computation community.

More in general, I find the introduction too generic (broad audience does not mean superficial readers, but specialists in different fields!). There are several mistakes in numbering the references that often do not correspond to the text: there is probably one reference missed with a consequent shift in numbering, but more citations seem to be chosen randomly.

Further points:

As concerns the spin Hamiltonian, the dipolar interaction between Tb-Tb magnetic moment can be easily estimated but values are not mentioned. In general, all the numerical values of the parameters used for estimating the energy levels should be clearly reported, at least in the supplementary information, in a way that positions estimated for LC can be reproduced.

The discussion on the individuation of the main (seven) LCs and spin levels is not fully convincing but it is quite understandable that the levels are not all are visible. Overall, I find the data analysis self-consistent and in line with what is done in the field.

Minor point: a color scale is probably missed in Fig4a

In summary, the weak points mentioned above must be clarified and relative discussion improved before considering this work suitable to be published.

Reviewer #3:

Remarks to the Author:

Biard et al. report on B-field controlled electron transport measurements through a back-gated Coulomb-blockaded single-molecule magnet (SMM). The SMM comprises a Tb₂Pc₃ complex with two S=3/2 nuclear spins, which are coupled indirectly via hyperfine coupling to the Tb electronic spins (J=6), which again have a ferromagnetic dipole-dipole (+exchange) coupling (C). The results are reported as electronic read-out of a coupled molecular qudit with d=16 nuclear spin states.

This work appears to be a continuation of an earlier paper by some of the same authors (Ref.43) in which this SMM was characterized. This, to my mind, is a very nice and systematic two stage process. Some of the same authors have earlier papers on a single (d=4) qudit (TbPc₂), in which they have demonstrated its qudit qualities and implemented Grover's algorithm, and as such this work on Tb₂Pc₃ clearly provides an important and highly non-trivial new development.

Overall, I consider this a very exciting and worthwhile development, which might be suitable for publication in Nat. Comm., depending on the reflections/answers the authors can provide to the questions and comments provided below.

1) As described in the supplement, the Coulomb blockaded molecule is measured at a chosen 'good' gate voltage near the charge degeneracy point at which the change in orientation of the electronic spin pair is visible as a B-field induced conductance switch. I cannot seem to find the chosen values, though? Where does it reside in Fig. S1b and what is the charge of the molecule at this V_G? A single cross is shown in Fig.S1b, but since a charging energy of 404 meV is somehow deduced, I presume that there is more data in which a diamond apex is seen and E_C can be read off? It would be instructive to get this overview before the zoom in to Fig.S1b is made. The cross itself is also a bit peculiar, and perhaps one can identify faint inelastic cotunneling lines on either side of it, hinting at incipient Kondo effects? I acknowledge the fact that this might be a very complicated story altogether and that not all details need to be understood for readers to accept the hyperfine aspect to the observed conductance switches reported here, but I am missing some information on these gross features in order to understand the physical setting for the SMM. Is V_G such that the Pc's are charged, say? If so, the electron on the Pc might interact with the Tb electrons. If practically no charge is transferred to the molecule, one of course needs not worry about this.

2) The operation of this system as a qudit has not been demonstrated, but one can surely imagine this as the next thing to do with this device, in line with their earlier realization for TbPc₂ (Grover algorithm). This means, however, that the title is slightly off. The main message of this work seems to be that quantum tunneling of magnetization (QTM), taking place from Landau-Zener

tunneling when sweeping B_X , gives a hysteretic switching of the conductance. This is beautifully demonstrated in Figs. 3a and 3c, and seems to confirm that this SMM maintains the hyperfine structure observed in Ref.43, even when placed between gold electrodes. The lifetime of the nuclear spin qudit, however, is found to be short compared to measurement time (1 ms), and therefore this device may not be functional as a coherent qudit to be operated faster than its decoherence/relaxation (here simply life-)time? Already the QTM reflected in the conductance is interesting, though, so perhaps the title ought to reflect that this is the finding, rather than tentatively promising this as a potential qudit? That aspect seems still to belong in the motivation and outlook.

3) How small is the estimated life time, and what current runs at zero bias to induce it?

4) Minor comment: In supplement, the phrase 'the presence of a quantum dot' is kind of strange. I would simply stick to Coulomb blockade. Quantum dots need not be Coulomb blockaded, and the conductance cross in the supplement is more complicated than a simple quantum dot.

5) The value of C appears to be taken from Ref.43, but one can easily imagine that this number changes when the molecule is placed between two gold electrodes. Could it be measured in situ from bias spectroscopy somehow? Fx. FM – AFM splitting measured from inelastic cotunneling, say? This bring me back to question 1 and the gross transport features of this device, and raises also the question of the nature of the two different conductance values between which the system flips when sweeping B_X . Is this a difference in cotunneling conductance for the two different spin configurations, or are we right at degeneracy, where one should think of sequential tunneling conductance? Or is there really a spin-blockade of sequential tunneling at the charge degeneracy point (couldn't tell with present resolution of Fig.S1b)?

We thank all referees for taking the time to carefully review our work and manuscript and for their valuable comments and suggestions. We have followed all the comments and made revisions in the manuscript and in the supplementary information accordingly which are highlighted in cyan. Our point-to-point responses to all questions and remarks are provided below. We hope that the referees will be satisfied with our revisions (highlighted in cyan in the MS and SI) and will agree that the manuscript is now acceptable for publication in *Nature Communications*. Thank you once more for your reviews and considerations.

Reviewer #1 (Remarks to the Author):

The manuscript entitled "Increasing the Hilbert space dimension using a coupled molecular spin qudit" reports on electrical transport measurements on a Tb₂Pc₃ molecule embedded in a transistor, as the first electronic read-out of two coupled qudits. The read-out procedure is based on the highly efficient detection of abrupt jumps in the differential conductance at the 'anti-crossing' field positions where QTM of the electronic spins occur.

One of the innovative aspects of the study is the demonstration through measurement, of the possibility to expand the Hilbert space of quantum objects by embedding dimers of qudits within the same molecule, here a Tb₂Pc₃ molecular magnet. The strategy essentially exploits the possibility of reading electronuclear transitions associated with weakly coupled lanthanide ions that carry magnetically active nuclear spins. The quadrupole term causes an uneven separation between hyperfine states, which together with the expansion of the Hilbert space, opens up new possibilities for individual qubit read-out and complex logic operations. From my point of view this paper is sufficiently innovative to be published in Nat. Commun. and will benefit the molecular magnetism community and others involved in quantum information research.

The authors probed unambiguously that interaction between nuclear and electronic states occur, and even quantified the dipolar coupling between qudits. Data agree well with the theoretical calculations and the experiments reported are appropriately described.

We thank the referee for his/her comments and greatly appreciate his/her opinion considering publication in Nature Communications.

Reviewer #2 (Remarks to the Author):

The manuscript of Biard et al. reports on proof-of-concept experiments that demonstrate the electrical readout of nuclear spin states of a prototypical molecular dimer. More specifically experiments were performed on a single "triple decker" Tb₂Pc₃ molecule embedded between electrodes in a transistor geometry. The key idea is novel and experiments are very challenging, thus results are unique and very interesting and they will certainly impact different fields and communities (molecular magnetism, quantum technologies, molecular electronics, spintronics etc.) and inspire several groups. In my opinion, this work merits publication in some way and it fits well the standard and audience of Nature Communication. Said that, I find two main weak points that need to be clarified and corrected.

We thank the referee for his/her comments and greatly appreciate his/her opinion about publication in Nature Communications. In the following, we answer to the different questions and points raised by the referee that needed to be clarified and corrected.

1. The magnetic field plays a key role in the whole work/manuscript but several discrepancies are evident comparing plots and reading the text. First, different orientations are not clearly defined: “ B_x is the direction of the coil (axis?) ...while B_y is applied along the other direction of the coil (?)” but to account for the Zeeman energy, B_z is introduced to indicate the molecular axis which is obviously randomly oriented with respect to the coil (x,y). Later (for instance in fig.1), B_z is also named as $B_{||}$. Secondly, the absolute value of the field spans different values: the level crossing is expected in (0-60)mT range (see fig.2, fig3a and c; fig.4 and 5) but a static field of about 1T is applied along x- and y- directions to bias the system to QTM regime. Authors honestly notice that: “... the experimental values do not correspond to the QTM position expected by eq.6...” and they also pointed out that in these experiments the system missed one orientation of the applied field, yet, as matter of fact, the presentation of results is not clear at all and there is probably something wrong.

The referee is correct and we apologize for discrepancies that are present in the manuscript when referring to the magnetic field. Indeed, the molecule is randomly oriented, thus we do not precisely know the orientation of the molecule with respect to the external applied magnetic field. For this reason, the whole investigation was conducted with respect to the axis of the coils (B_x , B_y). While discussing the Zeeman Hamiltonian we mistakenly introduced B_z as the field applied along the molecular easy axis. This is not correct, since we do not assess the molecular axis perpendicular to the plane of the coils (B_x , B_y). To clarify this, we have made the distinction in all the Figures (Fig. 1a-c, Fig. 3c and Fig. 4a), using $B_{||}$, as the molecular easy axis. Similarly, in all equations, we have replaced B_z by $B_{||}$ (as well as J_z and I_z by $J_{||}$ and $I_{||}$, to stay consistent). These changes are reflected in the manuscript and are highlighted in cyan. We thank the referee for his/her comment and hope that the manuscript will be easier to read now when referring to the magnetic fields compared to the molecule ($B_{||}$, as the molecular easy axis), and compared to our experimental set-up (coils B_x and B_y).

In the part “ Tb_2Pc_3 single-molecule magnet”, we obtain the Zeeman diagram for a zero-transverse field. It has to be noticed that the applied transverse field does not modify the anti-crossing position between them as regard to the applied field $B_{||}$; also, transverse fields do not modify the anti-crossing energy for a transverse field lower than 10 T. During the measurements, a static field B_y of about 1 T was applied to separate the signal emerging from the Tb_2Pc_3 molecular magnet from the anisotropic magnetic moments highlighted in Fig. 3b. If we had done the B_x magnetic field back and forth magnetic sweeps depicted in Fig. 3a at $B_y = 0T$, the signal would have been polluted and most probably screened by the magnetic noise. In order for the reader to better compare the measurements and the theory, we have mapped (centred) the B_x values on the central peak in the Fig. 3a,c,d, Fig. 4 and Fig. 5a. To clarify these points, we have added the following sentence in the caption in Fig. 3: “To ease the interpretation, the jump positions in the histograms have been centred at zero as regard to the central peak”.

2. From the very beginning (eg in the title title, abstract, introduction) as well as in the conclusions, the molecular spin system presented as promising qudit offering a large Hilbert space but it is not clear whether the authors look for a large multilevel quantum system or for

two entangled qudits. In the first case, no correlation (entanglement) is needed among the states (but only different energy spacing between levels) in the second case the entanglement between two distinguishable units play a key role. As a matter of fact, the Tb₂Pc₃ molecule presents two permanently coupled qudits but the entanglement between states is not discussed in this work. Although this does not affect much the results -which are focused only on the readout-, it sounds odd and even misleading for readers of Quantum Computation community.

We thank the referee for his/her comment as the point of our work is to experimentally demonstrate a multinuclear large-Hilbert-space compared to the one obtained for TbPc₂ for example. Along the whole discussion in the MS, we tried to stress the multi-levels characteristics of the studied single molecular system and the relevance of multilevel quantum bits for quantum computation. In this case, electronic coupling between two nuclear spin units will exponentially extend the Hilbert space available for quantum information processing. As the referee points out, a large Hilbert space and the uneven separation between states offer the possibility of realization complex quantum gates, which in turn can lead to the realization of complex quantum algorithms. Thus, we are definitely in the first case the referee points out: we are exploring and demonstrating a larger multilevel quantum system and not entangled qudits in this work. We clearly stated this point in the MS to not mislead the reader.

3. More in general, I find the introduction too generic (broad audience does not mean superficial readers, but specialists in different fields!). There are several mistakes in numbering the references that often do not correspond to the text: there is probably one reference missed with a consequent shift in numbering, but more citations seem to be chosen randomly.

We apologize for the errors found in the references. We have arranged the references as well as we have revised the introduction. Modifications are highlighted in cyan.

Further points:

4. As concerns the spin Hamiltonian, the dipolar interaction between Tb-Tb magnetic moment can be easily estimated but values are not mentioned. In general, all the numerical values of the parameters used for estimating the energy levels should be clearly reported, at least in the supplementary information, in a way that positions estimated for LC can be reproduced.

We thank the referee for this remark. We have now included the dipolar analysis in the SI, including all parameters for the simulation of the properties reported in the MS. The modifications in the SI are highlighted in cyan.

5. The discussion on the individuation of the main (seven) LCs and spin levels is not fully convincing but it is quite understandable that the levels are not all are visible. Overall, I find the data analysis self-consistent and in line with what is done in the field.

We thank the referee for appreciating the work and data analysis as the difficulties associated to the study. The calculation of the Hamiltonian in Eq 6 of the MS results in the Zeeman diagram presented in Figure 2.a. All intersections in the Zeeman diagram were carefully investigated and only 10 anticrossings were obtained from the calculations at the positions highlighted in Figure 2b. As at 3 of these positions the electronic spins reversal occurs at the

same magnetic field, only 7 positions in magnetic field where Quantum Tunneling of the Magnetization occurs could be experimentally resolved.

6. Minor point: a color scale is probably missed in Fig4a

We thank the referee for the observation. The colour scale has been added to Fig. 4a.

In summary, the weak points mentioned above must be clarified and relative discussion improved before considering this work suitable to be published.

We hope that our clarifications accompanying the revisions in the manuscript will be sufficient to make the referee considering our work and manuscript suitable for publication.

Reviewer #3 (Remarks to the Author):

Biard et al. report on B-field controlled electron transport measurements through a back-gated Coulomb-blockaded single-molecule magnet (SMM). The SMM comprises a Tb₂Pc₃ complex with two S=3/2 nuclear spins, which are coupled indirectly via hyperfine coupling to the Tb electronic spins (J=6), which again have a ferromagnetic dipole-dipole (+exchange) coupling (C). The results are reported as electronic read-out of a coupled molecular qudit with d=16 nuclear spin states.

This work appears to be a continuation of an earlier paper by some of the same authors (Ref.43) in which this SMM was characterized. This, to my mind, is a very nice and systematic two stage process. Some of the same authors have earlier papers on a single (d=4) qudit (TbPc₂), in which they have demonstrated its qudit qualities and implemented Grover's algorithm, and as such this work on Tb₂Pc₃ clearly provides an important and highly non-trivial new development.

Overall, I consider this a very exciting and worthwhile development, which might be suitable for publication in Nat. Comm., depending on the reflections/answers the authors can provide to the questions and comments provided below.

We thank the referee for his/her questions and comments. We provide in the following a detailed explanation of the questions raised by the referee.

1) As described in the supplement, the Coulomb blockaded molecule is measured at a chosen 'good' gate voltage near the charge degeneracy point at which the change in orientation of the electronic spin pair is visible as a B-field induced conductance switch. I cannot seem to find the chosen values, though? Where does it reside in Fig. S1b and what is the charge of the molecule at this V_G? A single cross is shown in Fig.S1b, but since a charging energy of 404 meV is somehow deduced, I presume that there is more data in which a diamond apex is seen and E_C can be read off? It would be instructive to get this overview before the zoom in to Fig.S1b is made. The cross itself is also a bit peculiar, and perhaps one can identify faint inelastic cotunneling lines on either side of it, hinting at incipient Kondo effects? I acknowledge the fact that this might be a very complicated story altogether and that not all details need to be understood for readers to accept the hyperfine aspect to the observed conductance switches reported here, but I am missing some information on

these gross features in order to understand the physical setting for the SMM. Is V_G such that the Pc's are charged, say? If so, the electron on the Pc might interact with the Tb electrons. If practically no charge is transferred to the molecule, one of course needs not worry about this.

We thank the referee for the questions and observations. The fact that we only depict one charge degeneracy point in Fig. S1 is to prevent the device from any damages. As a result, we limit the source drain voltage V_{SD} to low values ($<10\text{mV}$) in order to protect the molecular magnet integrity and its stability. Indeed, at higher V_{SD} , we already observed in other devices an increase of the noise signal or even a lost of the quantum dot signature as a higher current flowing through the molecule could irreversibly damage the transistor stability. Thus, we have not performed a Coulomb map for $V_{SD} > 10\text{mV}$.

However, as mentioned by the referee, an estimation of the charging energy is given. In Figure A1, we present measurements of the differential conductance and the current as a function of the gate voltage V_G . We clearly observe two peaks corresponding to two charge degeneracy points. If we consider that the peak observed at $V_G = -6\text{ V}$ corresponds to the same quantum dot than the one observed around $V_G = 0\text{ V}$, and from the slopes of the Coulomb diamond, we can obtain an estimation of the charging energy $E_C = 404\text{ meV}$. Note that due to the lack of additional measurements, this is only a rough estimation performed early in the study to verify that the observed quantum dot is compatible with the dimension of the molecular system. We did not performed any studies at $V_G = -6\text{ V}$ to prevent the gate dielectric from any leakage or even an irreversible damage of the molecular spin transistor. The measurements presented here and the discussion about the estimation of the charging energy are now included in the SI.

Figure A1. Measurement of the differential conductance (a) and the current (b) as a function of the gate potential V_G .

It is unlikely we can precisely answer where exactly the measurement point is located in the Coulomb Map depicted in Fig S1b compared to the Coulomb peak. However, the signatures of QTM through the measurements of abrupt jumps in the differential conductance were obtained on the right side of the charge degeneracy point, and this is now specified in the SI.

As mentioned by the referee, there is signatures of inelastic cotunneling. However, it goes in our opinion beyond the scope of the work we wanted to present in the manuscript, and we did not investigated these signatures, as the signal was weak.

We agree with the referee, if no charge is transferred to the molecule, we should worry about it. As in our previous works on TbPc₂, the current flows through the molecule via the Pc ligand. The charging energy of the Tb³⁺ ions is around 1eV [Zhu, European Journal of Inorganic Chemistry, 2004], thus it is very unlikely the quantum dot is one of the two Tb³⁺ ions, also because the QTM signatures would be totally different considering that the current can flow through the Tb³⁺ ions. The quantum dot is the Pc ligands which are known to be good electrical conductors due to their π delocalized links. Moreover, the overlap between the 4f shell of the Tb³⁺ ions and the π delocalized links of the Pc ligands give rise to an exchange coupling, as depicted in our previous work "Electrical single spin read-out using an exchange-coupled quantum dot" [Godfrin, ACS Nano, 2017]. Our detection scheme is then based on an exchange interaction between the electronic spin of the Tb³⁺ ions and the nearby Pc read-out quantum dot. The coupling between the two systems results in a spin-dependent conductance through the read-out dot and establishes an all electrical single spin detection⁺. The manuscript has been modified to specify this important point instead of only referring to the previous works.

2) The operation of this system as a qudit has not been demonstrated, but one can surely imagine this as the next thing to do with this device, in line with their earlier realization for TbPc₂ (Grover algorithm). This means, however, that the title is slightly off. The main message of this work seems to be that quantum tunneling of magnetization (QTM), taking place from Landau-Zener tunneling when sweeping B_X, gives a hysteretic switching of the conductance. This is beautifully demonstrated in Figs. 3a and 3c, and seems to confirm that this SMM maintains the hyperfine structure observed in Ref.43, even when placed between gold electrodes. The lifetime of the nuclear spin qudit, however, is found to be short compared to measurement time (1 ms), and therefore this device may not be functional as a coherent qudit to be operated faster than its decoherence/relaxation (here simply life-)time? Already the QTM reflected in the conductance is interesting, though, so perhaps the title ought to reflect that this is the finding, rather than tentatively promising this as a potential qudit? That aspect seems still to belong in the motivation and outlook.

A qubit is a 2-dimensional system; likewise a qudit is a d-dimensional system. However, we agree with the referee that if we have to demonstrate any coherent operations to name it a qudit, it is not possible as we only demonstrated so far an evidence of increasing the Hilbert space using a single molecular spin system. Our aim is certainly not to overclaim any results. However, this single device does not reflect a lifetime limitation for Tb₂Pc₃, and as mentioned by the referee, we are still working on to prove that this system could be used as a coherent qudit as our previous results on TbPc₂ clearly proved (see our answer to the next question) a lifetime and a coherence time functional to perform quantum algorithm.

We thus modify the title as "Increasing the Hilbert space dimension using a single coupled molecular spin" and refer in our manuscript and SI to multi-levels systems that might be promising as qudits.

3) How small is the estimated life time, and what current runs at zero bias to induce it?

We cannot give an exact value of the nuclear spin lifetimes. Due to our detection scheme (sweeping the external magnetic field), we can only determine that the lifetime of our nuclear

spins is shorter than the fastest magnetic field sweeping time. During our study, we compared the nuclear spin system effective temperature with or without a 15s waiting time between two magnetic field sweeps (300mT range at 100mT/s). We did not see any difference, indicating that the lifetime is shorter than the magnetic field sweeping time, that is 3s. In our previous studies on TbPc₂, we obtained different lifetimes depending on the devices. It was mainly due to the current flowing through the Pc ligands, and lifetimes from 34s to lower than 1s were obtained. We expect additional devices in the future exhibiting a larger lifetime, but we also work on our detection scheme to gain at least 3 orders of magnitude on our read-out time. The maximum current was about 3 nA at the charge degeneracy point centre and of the order of 0.1nA during the magnetic sweep measurements. Far from the degeneracy point, the current is of the order of the pA.

4) Minor comment: In supplement, the phrase ‘the presence of a quantum dot’ is kind of strange. I would simply stick to Coulomb blockade. Quantum dots need not be Coulomb blocked, and the conductance cross in the supplement is more complicated than a simple quantum dot.

We thank the referee for the suggestion. We have changed the “presence of a quantum dot” sentence to “signature of Coulomb blockade”. The change is highlighted in cyan.

5) The value of C appears to be taken from Ref.43, but one can easily imagine that this number changes when the molecule is placed between two gold electrodes. Could it be measured in situ from bias spectroscopy somehow? Fx. FM – AFM splitting measured from inelastic cotunneling, say? This bring me back to question 1 and the gross transport features of this device, and raises also the question of the nature of the two different conductance values between which the system flips when sweeping B_X. Is this a difference in cotunneling conductance for the two different spin configurations, or are we right at degeneracy, where one should think of sequential tunneling conductance? Or is there really a spin-blockade of sequential tunneling at the charge degeneracy point (couldn’t tell with present resolution of Fig.S1b)?

Yes, the value of C is taken from Moreno-Pineda, et al. (Inorg. Chem., 57, 9873 (2018)), and we agree with the referee that this value could be modified when the molecule is inserted in between two metallic contacts. However, we used this value in order to obtain an order of magnitude for the tunnel splitting at the different anti-crossings in order to confirm that at our magnetic field sweeping time, we only observed QTM and direct relaxations processes. As a result, we could not accessed to the FM to AFM transition from transport measurements, which was observed in the work of Moreno-Pineda, et al. (Inorg. Chem., 57, 9873 (2018)), using a micro-SQUID technic.

There is no spin-blockade in the used detection scheme. The working point is closed but not at the charge degeneracy point. This spin transistor can be split into two coupled quantum systems :

- (i) The 4f electrons of the two terbium Tb³⁺ ions for which only the FM configuration was accessible, so exhibiting a total electronic spin $J=12$ ($|\uparrow\uparrow\rangle$ or $|\downarrow\downarrow\rangle$).*
- (ii) the Pc ligands create the read-out quantum dot.*

This read-out quantum dot is tunnel-coupled to source and drain terminals to perform transport measurements. An overlap of the π -electron of the Pc ligands with the Tb³⁺ 4f

electrons gives rise to an exchange coupling between the read-out quantum dot and the electronic spin, without affecting its magnetic properties, as demonstrated by our measurements. As a result, the measured differential conductance through the read-out quantum dot will be function of the $|\uparrow\uparrow\rangle$ or $|\downarrow\downarrow\rangle$ configuration.

Reviewers' Comments:

Reviewer #2:

Remarks to the Author:

Authors have clarified open issues and revised the manuscript accordingly. the work is now improved and I recommend publication.

Reviewer #3:

Remarks to the Author:

Having considered the replies of the authors and their updated manuscript, I am happy to recommend it for publication.

REVIEWERS' COMMENTS

Reviewer #2 (Remarks to the Author):

Authors have clarified open issues and revised the manuscript accordingly. the work is now improved and I recommend publication.

Reviewer #3 (Remarks to the Author):

Having considered the replies of the authors and their updated manuscript, I am happy to recommend it for publication.

We thank the referees for taking the time to carefully review our work and manuscript and for their valuable comments and suggestions, and thank them for their recommendation to publish our revised manuscript in *Nature Communications*.